# Polydatin, a Glycoside of Resveratrol, Induces Apoptosis and Inhibits Metastasis Oral Squamous Cell Carcinoma Cells In Vitro

**DOI:** 10.3390/ph14090902

**Published:** 2021-09-05

**Authors:** Tae-Hyun Bang, Bong-Soo Park, Hae-Mi Kang, Jung-Han Kim, In-Ryoung Kim

**Affiliations:** 1Department of Oral Anatomy, School of Dentistry, Pusan National University, Busandaehak-ro, 49, Mulguem-eup, Yangsan-si 50612, Korea; bth2750@naver.com (T.-H.B.); parkbs@pusan.ac.kr (B.-S.P.); khaemi90@naver.com (H.-M.K.); 2Dental and Life Science Institute, School of Dentistry, Pusan National University, Yangsan 50612, Korea; 3Department of Oral and Maxillofacial Surgery, Medical Center, Dong-A University, Daesingongwon-ro, 26, Seo-gu, Busan 49201, Korea; omfsjhkim@dau.ac.kr

**Keywords:** polydatin, oral squamous cell carcinoma, apoptosis, autophagy, epithelial-mesenchymal transition

## Abstract

Although various methods, such as surgery and chemotherapy, are applied to the treatment of OSCC, there are problems, such as functional and aesthetic limitations of the mouth and face, drug side effects, and lymph node metastasis. Many researchers are making efforts to develop new therapeutic agents from plant-derived substances to overcome the side effects that occur in oral cancer treatment. Polydatin is known as a natural precursor of resveratrol, and research on its efficacy is being actively conducted recently. Therefore, we investigated whether polydatin can induce apoptosis and whether it affects cell migration and invasion through the regulation of EMT-related factors in OSCC. Polydatin decreased the survival and proliferation rates of CAL27 and Ca9-22 cells, and induced the release of cytochrome c, a factor related to apoptosis, and fragmentation of procaspase-3 and PARP. Another form of cell death, autophagy, was observed in polydatin-treated cells. In addition, polydatin inhibits cell migration and invasion, and it has been shown to occur through increased expression of E-cadherin, an EMT related factor, and decreased expression of N-cadherin and Slug and Snail proteins and genes. These findings suggest that polydatin is a potential oral cancer treatment.

## 1. Introduction

Cancer is the second leading cause of death in the world, and despite the many treatment methods being developed, the prevalence of cancer is still increasing, so it is considered a serious problem affecting the health of all human society [1]. In general, cancer can be classified into five types, depending on the type of tissue in which it occurs: carcinoma, sarcoma, leukemia, lymphoma, and myeloma. Among them, carcinoma refers to a malignant tumor originating from epithelial cells [2]. Oral squamous cell carcinoma (OSCC) is the most common oral cancer, a neoplasm that occurs in the oral cavity and pharynx, and it is known as the sixth-most common cancer worldwide due to various causes, such as modern people’s bad eating habits and smoking. The 5-year survival rate of OSCC patients is approximately 50%, and even after treatment with surgery and chemotherapy, many patients still suffer serious functional and aesthetic problems, recurrence, and metastasis [3,4]. Recently, cancer treatment has been rapidly developing, owing to the development of various anticancer drugs; despite this development, however, most cancers remain incurable, with a high mortality rate [5]. To overcome this difficult problem, many researchers are making great efforts to identify new anticancer substances, especially plant-derived chemicals, as potential candidate drugs without side effects [6].

Polydatin (resveratrol-3-O-β-mono-D-glucoside), a major component of *Polygonum cuspidatum*, is known as a glycoside of resveratrol [7]. Also, polydatin is usually found in products containing grapes, peanuts, hopcorns, and cocoa [7,8]. Studies have shown that polydatin displays antioxidant, anti-inflammatory, and anticancer properties. Moreover, it plays an important role in the treatment of cardiovascular, inflammatory, neurodegenerative, metabolic, and aging-related diseases [8]. 

Cell death is classified as either apoptosis or necrosis; in recent years, however, autophagy has been considered as another form of cell death, and it has been found to affect cell survival [9]. Unlike necrosis, apoptosis and autophagy are considered important mechanisms in anticancer research in that they do not cause an inflammatory response; in these processes, cells either commit suicide or are reused as an energy source for other cells [10,11]. The apoptosis–autophagy relationship is considerably complex, because such a relationship leads to different directions depending on the cell type and the stressor [10]. 

During normal embryonic and tissue development, changes in the shape and function of epithelial cells occur, resulting in a dynamic progression to mesenchymal stem cells, which is called epithelial–mesenchymal transition (EMT). In addition, EMT plays a significant role in tumor progression and metastasis, and research on EMT has been receiving attention recently [12]. When primary tumor cells proliferate, cancer cells metastasize after invading the bloodstream or the lymphatic system through the abnormal activation of the EMT, which is characterized by migration and invasiveness [13]. The ability of cells to invade and metastasize is a key feature of cancer progression, which involves a number of highly complex processes and various factors and signaling pathways. Metastasis via EMT through these complex processes and signaling pathways is considered an important issue in oral cancer, as in other carcinomas [14]. 

Therefore, in this study, to explore the potential of polydatin, a plant-derived compound, as an important therapeutic strategy for overcoming oral cancer, we aimed to investigate whether polydatin treatment induces apoptosis and blocks cell migration and invasion through EMT inhibition in oral squamous cell carcinoma cells, Ca9-22 and CAL27.

## 2. Results

### 2.1. Polydatin Inhibits the Viability and Proliferation of OSCC Cells 

We performed an MTT assay to determine the cytotoxicity by polydatin. Ca9-22 and CAL27 cells were treated with 0–2 mM polydatin and then incubated for 24–72 h to observe any changes in their viability (Figure 1A,B). To compare the difference in cytotoxicity of polydatin against normal cells and OSCC, cell viability was confirmed using keratinocytes, HaCaT, Ca9-22, and CAL27 cells. The IC50 value of HaCaT, a human keratinocyte, against polydatin was 2.37 mM, and the IC50 value of Ca9-22 and CAL27 cells were 1.15 and 0.97 mM, respectively. This result indicated that polydatin responds more sensitively to OSCC cells than keratinocytes (Figure 1C). Next, the effect of polydatin on cell proliferation was determined through a colony formation assay. Ca9-22 and CAL27 (3 × 10^2^) cells were treated with 0–1 mM polydatin; the number of colonies that formed after 7 days was determined and graphed. In both cell lines, the number of colonies that formed following treatments with up to 0.25 mM polydatin was nearly the same to that in the control group; however, the number of colonies rapidly reduced under treatment with at least 0.5 mM polydatin (Figure 1D,E). Therefore, polydatin inhibits the viability and proliferation of both OSCC cell lines

### 2.2. Polydatin Induces Apoptosis of OSCC Cells through the Mitochondrial Intrinsic Pathway 

To investigate whether polydatin, which is cytotoxic to OSCC, induces apoptosis, we first examined the nuclear morphology through Hoecst staining. Compared with the control, the Ca9-22 and CAL27 cells displayed nuclear condensation when treated with 1 mM polydatin, and they displayed a number of cleaved nuclei (i.e., apoptotic body formation) when treated with 1.5 mM polydatin (Figure 2). Based on these results, it was assumed that apoptosis was induced by 1 mM polydatin; immunofluorescence experiments were thus performed using 1 mM polydatin, whereas Western blot analyses were performed using 0, 0.25, 0.5, 0.75, and 1 mM polydatin. The immunofluorescence assay results showed that polydatin released the cytochrome c protein from the mitochondria of the Ca9-22 and CAL27 (Figure 3A) cells. When the apoptosis program is activated, the activation of caspase proteins is particularly accompanied. The proteins that act as the main switches for activation include Bcl-2 and bcl-xl mcl-1, which regulate cell survival, whereas Bax and Bak, instead, promote cell death [15]. Meanwhile, the Western blot results (Figure 3B) showed that polydatin decreased bcl-2 expression and increased bax in both cell lines dose-dependently. In addition, the expressed proteins were activated in the final stage of apoptosis of the Ca9-22 and CAL27 cells (Figure 3C,D); that is, treatment with 1 mM polydatin induced the activation of caspase protein, the cleavage of procaspase-3, and the fragmentation of PARP in both cell lines. These results suggested that treatment with 1 mM polydatin induces apoptosis in OSCC cells. 

### 2.3. Polydatin Induces Autophagy in OSCC Cells

We confirmed the formation of autophagosomes and the expression of autophagy-related proteins ATG5 and LC3 using AO and MDC staining and Western blot analysis. The MDC staining results showed that the Ca9-22 and CAL27 cells treated with 1 mM polydatin formed autophagosomes (Figure 4A), whereas AO staining revealed the formation of AVOs, which stained red (Figure 4B). In both cell lines, treatment with 0.25 mM polydatin or higher increased the expression levels of ATG5 and LC3 (Figure 4C), indicating that polydatin induces autophagy in OSCC cells.

### 2.4. Polydatin Prevents EMT in OSCC Cells

Finally, we examined the changes in cell migration, invasion, and expression of EMT-related proteins to determine whether polydatin possesses anti-metastatic properties. In the wound healing assay, treatment with 0.25 and 0.5 mM polydatin inhibited the migration of Ca9-22 and CAL27 cells (Figure 5A,B). In an invasion assay, the transwell chamber is coated with Matrigel to determine whether cells could penetrate the Matrigel and reach the membrane of the transwell within 72 h. In the present experiment, it was confirmed that 0.5 mM polydatin significantly decreased the penetration of the Ca9-22 and CAL27 cells (Figure 5C,D). Meanwhile, immunofluorescence assay was employed to determine how polydatin alters the expression of E-cadherin, an adhesion molecule between cells. Treatment with 0.5 mM polydatin increased the expression of E-cadherin at cell junctions in both cell lines (Figure 6A). E-cadherin, an epidermal factor, increased the expression levels of proteins and genes in the group treated with 0.5 mM polydatin or higher. N-cadherin, Snail, and Slug (mesenchymal factors) expression levels were decreased in the group treated with 0.25 mM polydatin or higher (Figure 6B–D). These results indicated that polydatin exerts an antimetastatic effect toward OSCC cells through the EMT process. 

## 3. Discussion

### 3.1. Toxicity of Polydatin for Cancers

While various anticancer drugs employed in chemotherapy induce cancer cell apoptosis, they cause excessive damage to normal cells and thus elicit serious side effects [16]. Additionally, cancer recurrence or metastasis, which threaten the survival of patients, remains a challenge [17]. OSCC is the most common oral cancer, accounting for more than 90% of oral malignancies, and it has a poor prognosis that is associated with local recurrence and distant metastasis due to resistance to chemotherapy [18]. Given the pharmacological effects of many plant-derived substances, herbal medicinal remedies have been used for centuries [19]. The use of some of these substances is an emerging cancer treatment strategy in modern medicine, as their efficacy in inhibiting cell division and metastasis and in inducing apoptosis has been proven [20]. Polydatin is a natural precursor of resveratrol, and studies have already reported on its biological functions, such as inhibition of inflammation, immune regulation, induction of tumor cell apoptosis, and provision of protection against liver damage and cardiovascular disease [7]. Thus far, studies on the anticancer activity of polydatin have shown the following: NF-κB suppressed by polydatin, inhibited NLRP3 inflammasome activation and it lead to prevent proliferation and metastasis of non-small cell lung cancer cells [21]. In colorectal cancer, the antitumor effects of polydatin was seen in miR-382/PD-L1 regulation [22]. Additionally, a combination treatment involving polydatin and 2-deoxy-d-glucose in breast cancer induced apoptosis by targeting ROS/PI3K/AKT/HIF-1α/HK2 [23]. However, studies on polydatin-induced apoptosis and metastasis in OSCC cells remain scarce. Therefore, we investigated the anticancer effects of polydatin by using the oral cancer cell lines Ca9-22 and CAL27. 

Studies have found that the viability of MCF7 (a breast cancer cell line), HepG2 (a hepatocellular carcinoma), and MG-63 (a human osteosarcoma) reached 50% following treatment with approximately 100 µM polydatin [23,24,25]. Polydatin inhibited the protein expressions of AHR, CYP1A1, and HSP-90, which are tumor grade-related markers, and the IC50 concentration of polydatin was 20 μM [26]. Also, it was reported that polydatin (IC50 value was 20 μM) inhibits cell cycle and survival by inhibiting the accumulation of reactive oxygen species and a strong increase in ER stress in CAL27 [27]. However, these studies did not provide clear evidence for apoptosis (nuclear condensation, DNA fragmentation, and caspase expression, etc.). In the present study, polydatin induced apoptosis at a concentration that is 10 times higher than that used in many studies. Whether this phenomenon occurs only in OSCC warrants further investigations. In this study, polydatin demonstrated a high cytotoxicity at a concentration of 1 mM or higher, and it clearly inhibited cell proliferation at a concentration of 0.5 mM or higher (Figure 1).

### 3.2. Induction of Apoptosis by Polydatin

Apoptosis is a mechanism leading to suicide in response to DNA damage and is an important process in various biological systems, including maintenance of cellular homeostasis, embryonic development, and immune response to infection [28]. Therefore, inducing apoptosis in cancer cells has become the basis for targeted therapies with minimized adverse effects toward normal cells [29]. When a cell enters the intrinsic pathway of apoptosis, the proteins present in the mitochondria are changed, and in particular, proteins such as cytochrome c are released into the cytoplasm to form a complex with the caspase protein present in the cytoplasm to undergo DNA fragmentation [30]. Our results showed that polydatin induced the activation of caspase through the release of cytochrome c, a protein present in mitochondria; subsequently, PARP fragmentation occurred, leading to a fragmented nucleus called an apoptotic body, which was observed in Ca9-22 and CAL27 cells (Figure 2 and Figure 3). Thus, polydatin induces apoptosis through the mitochondrial endogenous pathway in OSCC cells.

### 3.3. Inhibiton of EMT by Polydatin

Local or distant metastasis of OSCC is a major cause of adverse effects on the survival of OSCC patients [31]. EMT is the process by which epithelial cells differentiate into cells that display the characteristics of mesenchymal cells; intercellular infiltration and migration subsequently occur, leading to a normal embryonic development or to cancer metastasis [32]. In epithelial cells, when E-cadherin is inhibited, cell–cell junctions dissociate and take on the characteristics of mesenchymal cells, resulting in EMT [33]. EMT is characterized by N-cadherin upregulation followed by E-cadherin downregulation, a process controlled by a complex network of signaling pathways and transcription factors [34]. Our study confirmed that polydatin upregulates the protein and gene expression levels of E-cadherin, thereby inhibiting cell migration and invasion. Also, the mesenchymal marker N-cadherin was downregulated by polydatin treatment. Snail, Slug, Zeb1/2, and Twist1/2, which are EMT transcription factors, are known to regulate the EMT process [35]. The expression of Snail and Slug downregulates E-cadherin expression and upregulates vimentin and fibronectin expression, leading to the development of a complete EMT phenotype [36]. Our results showed that Snail and Slug were downregulated by polydatin treatment. Therefore, polydatin upregulates E-cadherin and downregulates N-cadherin through the suppression of Snail and Slug, thereby inhibiting the migration and invasion of oral cancer cells (Figure 6). 

### 3.4. Proposal of a Therapeutic Approach Using Polydatin in OSCC

As demonstrated in several studies, there is no doubt that polydatin has strong anticancer activity. However, it is considered that a more in-depth study is needed to elucidate the difference in the activity of polydatin according to cell types. Nanotechnology is regarded as an innovative strategy to combat cancer, and silver and gold nanobiomaterials have been shown to have significant potential for targeted drug delivery systems [37]. Plant-mediated biomimetic synthetic silver nanoparticles (AgNPs) are known to be less toxic and more environmentally friendly than chemically synthesized nanoparticles. Recently, natural sources, including plants and microorganisms, have synthesized AgNPs mediated and demonstrated that they are effective against lung and cervical cancer cells [2,38]. Also, a study to increase cell membrane permeability by encapsulating polydatin was published. They found that polydatin and metformin encapsulated with Poly[lactic-co-glycolic acid] (POL-PLGA-NPs) were to induce a mitochondrial-mediated apoptotic mechanism in vivo [39]. Therefore, in our results, the reason that a rather high concentration of polydatin induced apoptosis and inhibited metastasis could be inferred that polydatin was not properly absorbed during penetration into the cell membrane. However, since this study was conducted in vitro, it is considered that additional animal experiments are needed to confirm the above.

Taken together, polydatin induces apoptosis through the mitochondria of oral cancer cells and inhibits cell migration and invasion by regulating the EMT-related transcription factors. These results suggested that polydatin is a potential anticancer agent capable of inhibiting the metastasis of OSCC. However, since this phenomenon occurred at a high concentration of polydatin, it is necessary to study the ability of polydatin to increase the penetration into OSCC, and further, its efficacy should be revealed through animal experiments. If these things are supported, polydatin could be a novel therapeutic agent for various type of cancer, as well as OSCC.

## 4. Materials and Methods

### 4.1. Cell Culture and Polydatin Treatment

The human OSCC cell lines CAL27 and Ca9-22 were purchased from the American Type Culture Collection (ATCC), and the human keratinocyte HaCaT was provided by Professor Young-Hyeon Yoo, Dong-A University (Busan, Korea). Both cells were cultured in Dulbecco’s Modified Eagle’s Medium (DMEM, Hyclone, Logan, UT, USA) medium containing 10% fetal bovine serum (FBS, Hyclone, Logan, UT, USA), and 1% antibiotics under 5% CO_2_ at 37 °C incubator. Cells from 15 to 20 passages were used and subcultured once every 2–3 days. A 100 mM stock solution was prepared by dissolving polydatin in DMSO and diluting it to the required concentration.

### 4.2. MTT Assay 

HaCaT, CAL27 and Ca9-22 (1 × 10^4^) cells were seeded in a 96-well plate and incubated in an incubator for 24 h to allow the cells to attach to the well surface. Both cell lines were subsequently treated with various polydatin concentrations and then cultured for 24–72 h. After the medium was removed, each well was treated with MTT solution and incubated at 37 °C for 4 h. Subsequently, the supernatant was removed, the formazan crystals were dissolved using DMSO, and the absorbance was measured at 570 nm using a SpectraMax iD3 microreader (BioTek, Winooski, VT, USA). Absorbance was measured in three independent experiments, and the results are expressed as mean ± standard deviation (SD).

### 4.3. Colony Formation Assay

CAL27 and Ca9-22 (3 × 10^2^) cells were seeded in a six-well plate and cultured for 24 h to allow the cells to adhere to the well surface. Each well was subsequently treated with polydatin and cultured for 7 days. After the supernatant was removed, each well was fixed with methanol for 15 min, and the cell colonies that formed in the wells were stained with 0.5% crystal violet dye for 30 min. After the cells were washed three times with tap water and sufficiently dried at room temperature, the entire plate was photographed, and the colony units were counted.

### 4.4. Hoechst 33342 Staining

CAL27 and Ca9-22 cells were seeded in a 24-well plate and cultured for 24 h to allow the cells to attach to the well surface. Both cell lines were treated with polydatin and cultured for 24 h, and then stained with Hoechst 33,324 for 10 min. The cells’ nuclear morphology was photographed at 100× using a Lionheart FX Automated Microscope (BioTek, Winooski, VT, USA). The number of normal nuclei and abnormally condensed nuclei was determined and graphed. Data were collected from three independent experiments, and each number is expressed as mean ± SD.

### 4.5. Immunofluorescence Staining

CAL27 and Ca9-22 cells were seeded into a Lap-Tek chamber slide and cultured for 24 h to allow the cells to attach to the well surface. The next day, both cell lines were treated with 1 mM polydatin and cultured for 24 h. The cells were immediately washed three times with PBS, fixed with 4% PFA for 15 min, and again washed three times with PBS. For the antibody permeation, the cells were treated with 0.2% Triton-X 100 (in PBS) solution for 10 min. The primary antibody (cytochrome c, cleaved procaspase3, Ecadharin; Cell Signaling Technology (Beverly, MA, USA), was diluted at 1:100 in 1% BSA (in PBS) solution and applied to the cells for 24 h at 4 °C. Thereafter, the cells were washed three times with PBS, and the FITC-conjugated secondary antibody (Enzo Life Sciences, Farmingdale, NY, USA) was applied at a ratio of 1:100 on the cells, reacted at room temperature for 2 h, and the process of washing with PBS was repeated. The nuclei, actin, and mitochondria were stained with DAPI, rhodamine phalloidin, and mitotracker (Invitrogen (Gaithersburg, MD, USA), respectively. Each stained sample was observed, photographed, and analyzed with a Zeiss LSM 750 laser-scanning confocal microscope (G €oettingen, Germany).

### 4.6. Arcridin Orange (AO) and Monodansylcadaverine (MDC) Staining

CAL27 and Ca9-22 cells were seeded into a Lap-Tek chamber plate and cultured for 24 h to allow the cells to attach to the well surface. The next day, both cell lines were treated with 1 mM polydatin and cultured for 24 h. Each cell was stained with AO at room temperature for 5 min and then immediately observed, photographed, and analyzed using a Lionheart FX Automated Microscope (BioTek, Winooski, USA). MDC staining was performed similar to AO staining, and the cells were photographed and analyzed using a Lionheart FX Automated Microscope. 

### 4.7. Wound-Healing Assay 

CAL27 and Ca9-22 cells were seeded approximately 70–80% confluently in a six-well plate and cultured for 24 h to allow the cells to adhere to the well surface. Cells were then starved for 24 h in serum-free medium supplemented with 0.2% BSA. The bottom of the six-well plate was scratched using a yellow tip. The cells were treated with 0, 0.25, and 0.5 mM polydatin in complete medium and fixed with 4% PFA after 24 h. The cells were stained with Hoechst stain and then observed, photographed, and analyzed using a Lionheart FX Automated Microscope.

### 4.8. Cell Invasion Assay 

A trans-well membrane (Corning Costar, Cambridge, MA, USA) was coated with 30 μL Matrigel^®^ and incubated for 4 h and then CAL27 and Ca9-22 cells were seeded. Subsequently, the cells were treated with 0.5 mM polydatin and cultured for 48 h. The medium applied to the upper chamber of the Matrigel^®^-coated trans-well was a serum-free medium containing 0.5 mM polydatin, and the medium applied to the lower chamber was filled with a medium containing 10% FBS, but not polydatin. The cells in the upper chamber were washed with PBS and fixed with 100% methanol for 10 min. The cells were stained with hematoxylin–eosin (H&E) for 30 min. After the membrane in the upper chamber was separated, it was washed three times in PBS, dehydrated through immersion in 70%, 80%, 90%, and 100% ethanol, mounted using malinol, and then photographed and analyzed using a Lionheart FX Automated Microscope. In the total area, the area of cells that penetrated the tran-swell and invaded on the membrane was digitized with Adobe Photoshop CS (San Jose, CA, USA) and expressed as a graph.

### 4.9. Western Blot Assay

Polydatin-treated CAL27 and Ca9-22 cells were harvested and lysed using RIPA buffer at 4 °C for 2 h. Proteins were quantified using Bradford assay (Bio-Rad, Richmond, CA, USA), and the amount of protein in each sample was set to 20 µg. For the electrophoresis, 10% SDS-PAGE gel was used, and the samples were run at 100 V and 20 mA. The protein was transferred onto a PVDF membrane (Millipore, Billerica, MA) for more than 16 h at 20 V. The membrane was blocked for 1 h with 5% non-fat dry milk, and the primary antibody was applied at a 1:1000 ratio and incubated at room temperature for 1 h. The membrane was washed five times with PBS for 10 min each round, and the secondary antibody was applied at 1:5000 and incubated at room temperature for 1 h. The membrane was washed five times with PBS for 10 min each round and incubated with SuperSignal West Femto enhanced chemiluminescence substrate; protein expression was detected with ImageQuant LAS 500 chemiluminescence (GE Healthcare, Chicago, IL, USA).

### 4.10. RNA Isolation and RT-qPCR 

Total RNA was purified from polydatin-treated CAL27 and Ca9-22 (4 × 10^5^ cells/well) cells using RNeasy mini kit (Qiagen Inc., Valencia, CA, USA), and RNA quantification was using Nano-Drap (Thermo Scientific, NanoDrop 2000 Spectophotometer, USA). The total amount of RNA was set to 2 μg, and cDNA was synthesized using RevertAid First-Strand cDNA Synthesis Kit (Thermo Fisher Scientific, Pittsburgh, PA, USA). The cDNA synthesis was performed according to the manual provided by the manufacturer and as previously described [40]. Each sample for quantitative PCR includes primers of each gene, DEPC water, cDNA and 2× SYBR qPCR Mix (Applied Biosystems, Warrington, UK), and the gene was amplificated on the qPCR Detection System (Applied Biosystems, ABI 7500, Foster City, CA, USA). The sequence of each gene primer is shown as below; E-cadherin, forward: AGTGACTGATGCTGATGCCC, reverse: AATGTACTGCTGCTTGGCCT; N-cadherin, forward: TTGCCAGAAAACTCCAGGGG, reverse: TGGCCCAGTTACACGTATCC; Snail, forward: GTTTACCTTCCAGCAGCCCT, reverse: TCCCAGATGAGCATTGGCAG; and Slug, forward: GCTACCCAATGGCCTCTCTC, reverse: CTTCAATGGCATGGGGGTCT. Relative mRNA expression levels for genes were analyzed using the 2^–∆∆Cq^ method [41].

### 4.11. Statistical Analysis

All data are expressed as mean ± SD. The figures presented herein were derived from at least three independent experiments. Statistical analysis was performed using one-way analysis of variance (ANOVA) and Dunnett’s comparison. Differences with a probability (*p*) value of less than 0.05 were considered statistically significant.

## Figures and Tables

**Figure 1 pharmaceuticals-14-00902-f001:**
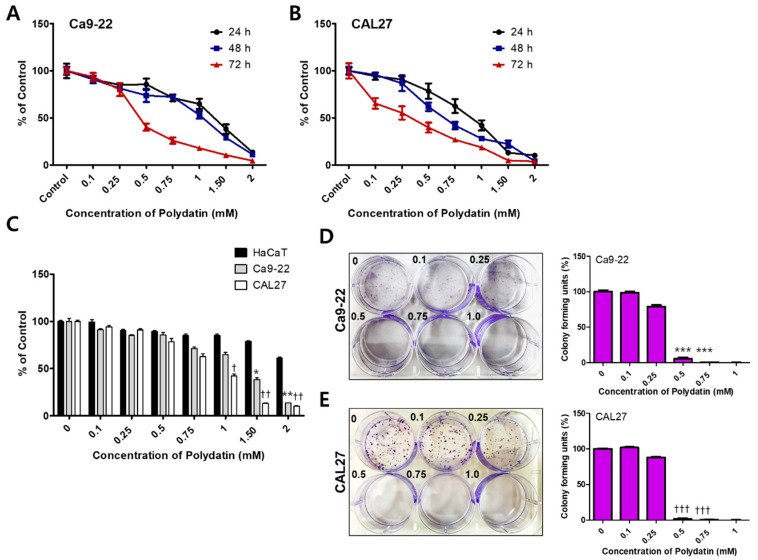
Inhibitory effect of polydatin on OSCC cell viability and proliferation. Ca9-22 (**A**) and CAL27 (**B**) cells were treated with 0–2 mM polydatin for 24–72 h and then their viability was assessed. (**C**) Comparison of cell viability in human keratinocyte HaCaT and OSCC (Ca9-22 and CAL27) cells by polydatin treatment for 24 h. Ca9-22 (**D**) CAL27(**E**) cells were treated with 0–1 mM polydatin for 7 days. The Ca9-22 (**D**) and CAL27(**E**) cell colonies that formed were stained with 1% crystal violet and then photographed with a digital camera. The number of colonies that formed in each group was counted and graphed. Graphs showing the formation rate of each colony, with the control group representing a 100% formation rate. Results are expressed as mean ± SD. Data were derived from three independent experiments. (Ca9-22: * *p* < 0.05, ** *p* < 0.01, *** *p* < 0.001; CAL27: † *p* < 0.05, †† *p* < 0.01, ††† *p* < 0.001).

**Figure 2 pharmaceuticals-14-00902-f002:**
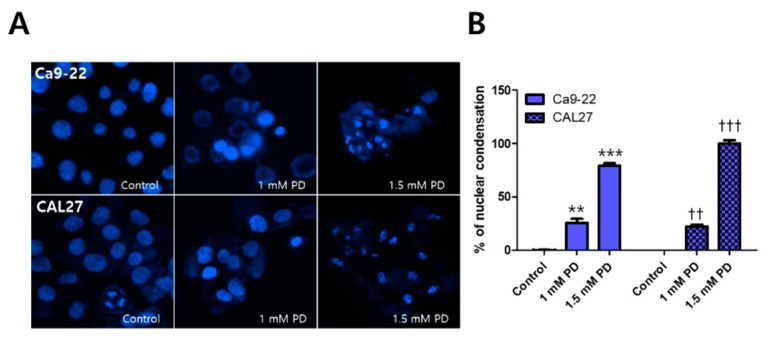
Polydatin-induced changes in nuclear morphology in OSCC cells. Ca9-22 and CAL27 cells were treated with 0, 1, and 1.5 mM polydatin. The nuclei morphology in both cell lines was observed through Hoechst staining. (**A**) The nuclear morphologies of Ca9-22 and CAL27 were observed and photographed under a fluorescence microscope. (**B**) The number of cells with a condensed nucleus is presented in a histogram. Results are expressed as mean ± SD. Data were derived from three independent experiments. (Ca9-22: ** *p* < 0.01, *** *p* < 0.001; CAL27: †† *p* < 0.01, ††† *p* < 0.001).

**Figure 3 pharmaceuticals-14-00902-f003:**
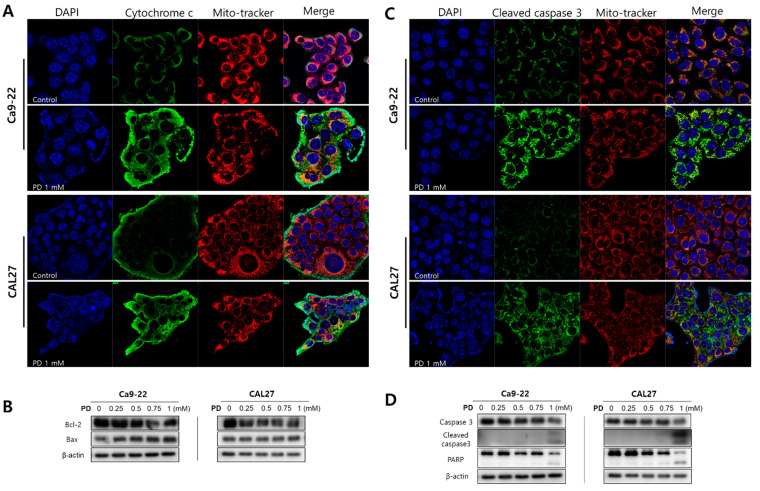
Expression of apoptosis-related proteins by polydatin in OSCC cells. After a 24-h treatment with 1 mM polydatin, the expression of (**A**) cytochrome c and (**C**) cleaved caspase-3 in Ca9-22 and CAL27 cells were confirmed through immunofluorescence staining as observed under a confocal microscope. After a 24-h treatment with 0–1 mM polydatin, the protein expression of (**B**) bcl-2, bax (**D**) caspase-3, cleaved caspase-3, and PARP was confirmed through Western blot analysis. β-actin was used as loading control.

**Figure 4 pharmaceuticals-14-00902-f004:**
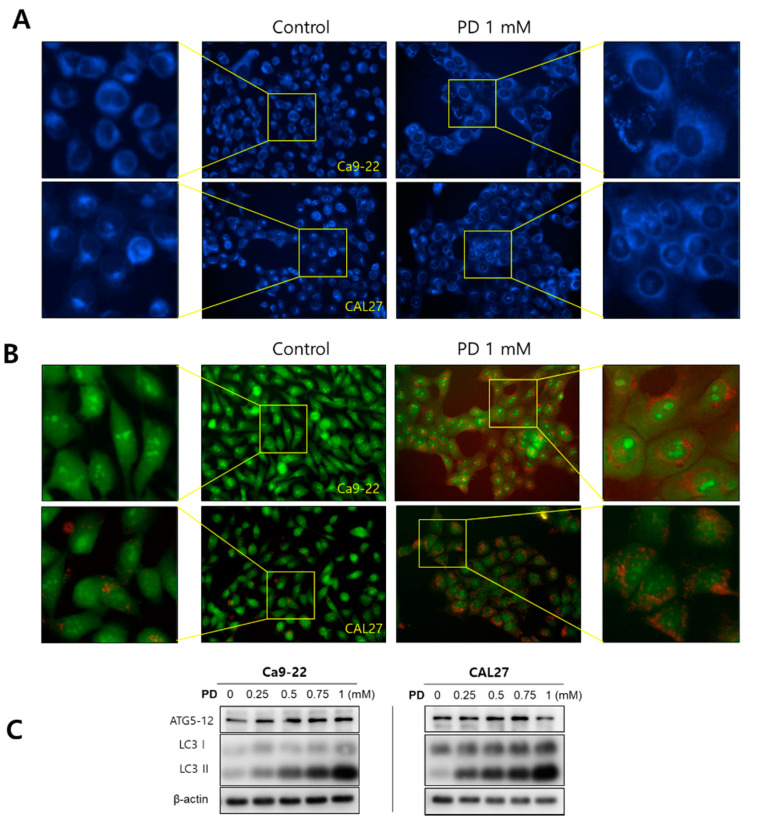
Induction of autophagy by polydatin in OSCC cells. (**A**) MDC staining and (**B**) AO staining were performed to detect autophagosomes and AVOs. (**C**) Ca9-22 and CAL27 cells were treated with 0–1 mM polydatin, and the expression of the autophagy-related proteins Atg5 and LC3B was confirmed. β-actin was used as loading control.

**Figure 5 pharmaceuticals-14-00902-f005:**
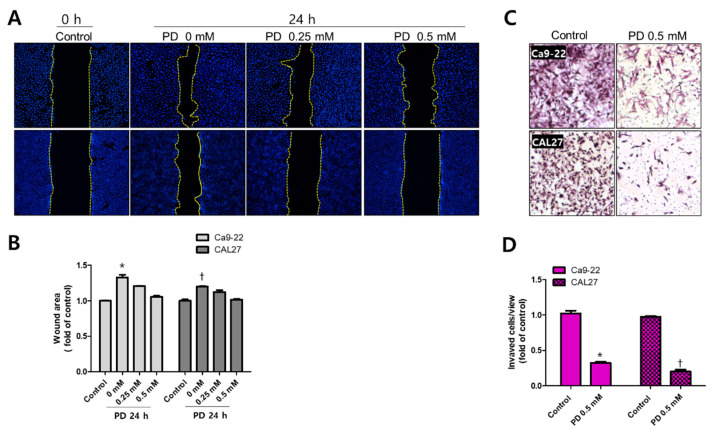
Inhibition of cell migration and invasion by polydatin in OSCC cells. (**A**) Ca9-22 and CAL27 cells seeded in a six-well plate were scratched and then treated with 0, 0.25, and 0.5 mM polydatin; the cell migration distance was measured through Hoechst staining 24 h later. (**B**) The wound healing rates in the two cell lines were measured and graphed. Wound width was calculated and analyzed using the ImageJ software. (**C**) Ca9-22 and CAL27 cells were seeded in a transwell membrane and treated with 0.5 mM polydatin for 48 h; the invading cells were stained with H&E. (**D**) The number of invading cells was determined and graphed. Results are expressed as mean ± SD. Data were derived from three independent experiments. (Ca9-22: * *p* < 0.05; CAL27: † *p* < 0.05).

**Figure 6 pharmaceuticals-14-00902-f006:**
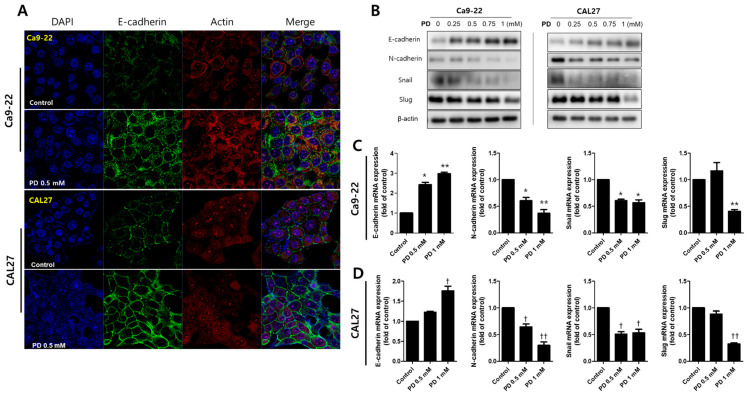
Polydatin-regulated expression of EMT-related proteins and genes in OSCC cells. (**A**) Ca9-22 and CAL27 cells were treated with 0.5 mM polydatin for 24 h, subjected to immunofluorescence assay, and imaged and analyzed by confocal microscopy. The nuclei, E-cadherin, and actin were stained with DAPI (blue), Alexa 488 (green), and rhodamine phalloidin (red), respectively. (**B**) Ca9-22 and CAL27 cells were treated with 0–1 mM polydatin, and their expression levels of autophagy-related proteins, namely, E-cadherin, N-cadherin, Snail, and Slug, were determined. β-actin protein was used as internal control. (**C**) Ca9-22 and (**D**) CAL27 cells were treated with 0, 0.5, and 1 mM polydatin for 24 h, and the gene expression levels of E-cadherin, N-cadherin, Snail, and Slug were determined through QPCR. Results are expressed as mean ± SD. Data were derived from three independent experiments. (Ca9-22: * *p* < 0.05, ** *p* < 0.01; CAL27: † *p* < 0.05, †† *p* < 0.01).

## Data Availability

The data presented in this study are available on request from the corresponding author.

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
