# Peer review of "Polydatin, a Glycoside of Resveratrol, Induces Apoptosis and Inhibits Metastasis Oral Squamous Cell Carcinoma Cells In Vitro"

_pharmaceuticals, 2021, doi:10.3390/ph14090902_

Round 1
Reviewer 1 Report
Thank you for your efforts . Only absorption and pharmacokinetic study and data are required.
What is the main question addressed by the research?
Researcher showed that polydatin is a potential anticancer agent capable of inhibiting the metastasis of OSCC though it needs a high concentration and data for pharmacokinetics is necessary along with animal model study as I mentioned in the review report.
Is it relevant and interesting?
Yes, fairly interesting
How original is the topic?
Fair
What does it add to the subject area compared with other published material?
The manuscript focused on potentiality of polydatin for OSCC, though it is already a reported anticancer agent for other cancer cell lines.
Is the paper well written?
Fair
Is the text clear and easy to read?
Fair, but can be improved
Are the conclusions consistent with the evidence and arguments presented?
Yes
Do they address the main question posed?
Yes, but can be improved with the experiment I have already suggested.
Author Response
Response to review report
Report 1
Thank you for your efforts Only absorption and pharmacokinetic study and data are required.
What is the main question addressed by the research? Researcher showed that polydatin is a potential anticancer agent capable of inhibiting the metastasis of OSCC though it needs a high concentration and data for pharmacokinetics is necessary along with animal model study as I mentioned in the review report.
Is it relevant and interesting? Yes, fairly interesting
How original is the topic? Fair
What does it add to the subject area compared with other published material? The manuscript focused on potentiality of polydatin for OSCC, though it is already a reported anticancer agent for other cancer cell lines.
- Thank you for your valuable comments. As you said, studies on the effects of polydatin on other cancer cells have already been published or are ongoing. However, studies on the anticancer activity of polydatin in oral squamous cell carcinoma cells are still limited. In this study, unlike the effects of polydatin in other carcinomas, high concentrations of polydatin showed effects such as induction of apoptosis and inhibition of cell migration and invasion in oral squamous cell carcinoma. In the discussion, we inferred that polydatin was not properly absorbed during the cell membrane penetration process. Thereby, we have supplemented at the discussion section in this manuscript as follows by citing few references. In addition, we cited and supplemented a few references in the discussion section as follows.;
- As demonstrated in several studies, there is no doubt that polydatin has strong anticancer activity. However, it is considered that a more in-depth study is needed to elucidate the difference in the activity of polydatin according to cell types. Nanotechnology is regarded as an innovative strategy to combat cancer, and silver and gold nanobiomaterials have been shown to have significant potential for targeted drug delivery systems [38]. Plant-mediated biomimetic synthetic silver nanoparticles (AgNPs) are known to be less toxic and more environmentally friendly than chemically synthesized nanoparticles. Recently, natural sources, including plants and microorganisms, have synthesized AgNPs mediated and demonstrated that they are effective against lung and cervical cancer cells [2,39]. Also, a study to increase cell membrane permeability by encapsulating polydatin was published. They found that polydatin and metformin encapsulated with Poly[lactic-co-glycolic acid] (POL-PLGA-NPs) were to induce a mitochondrial-mediated apoptotic mechanism in vivo [40]. Therefore, in our results, the reason that a rather high concentration of polydatin induced apoptosis and inhibited metastasis could be inferred that polydatin was not properly absorbed during penetration into the cell membrane. However, since this study was conducted in vitro, it is considered that additional animal experiments are needed to confirm the above.
- Barabadi, H.; Hosseini, O.; Damavandi Kamali, K.; Jazayeri Shoushtari, F.; Rashedi, M.; Haghi-Aminjan, H.; Saravanan, M. Emerging Theranostic Silver Nanomaterials to Combat Lung Cancer: A Systematic Review. Journal of Cluster Science 2020, 31, 1-10, doi:10.1007/s10876-019-01639-z.
- Saravanan, M.; Barabadi, H.; Vahidi, H.; Webster, T.J.; Medina-Cruz, D.; Mostafavi, E.; Vernet-Crua, A.; Cholula-Diaz, J.L.; Periakaruppan, P. Chapter 19 - Emerging theranostic silver and gold nanobiomaterials for breast cancer: Present status and future prospects. In Handbook on Nanobiomaterials for Therapeutics and Diagnostic Applications, Anand, K., Saravanan, M., Chandrasekaran, B., Kanchi, S., Jeeva Panchu, S., Chen, Q., Eds.; Elsevier: 2021; pp. 439-456.
- Barabadi, H.; Vahidi, H.; Kamali, K.D.; Rashedi, M.; Saravanan, M. Antineoplastic biogenic silver nanomaterials to combat cervical cancer: a novel approach in cancer therapeutics. Journal of Cluster Science 2020, 31, 659-672.
Is the paper well written? Fair
Is the text clear and easy to read? Fair, but can be improved
Are the conclusions consistent with the evidence and arguments presented? Yes
Do they address the main question posed? Yes, but can be improved with the experiment I have already suggested.
- We appreciate your valuable interest and comments, and we look forward to publishing this paper in the 'Pharmaceuticals'.

Reviewer 2 Report
The manuscript shows polydatin could be potential drug for the treatment of OSCC. Through a series of experiments, the authors present PD could inhibit cell proliferation, invasion, and metastasis by inducing apoptosis and autophagy and increasing E-Cadherin expression.
However, we still need to note:
- The findings in the works lacks significance, because PD has not only tested for the potential use for other types of cancer, but also the concentration of PD used in the paper is very high, making it hard for in vivo application.
- In figure 3A, the mito-traker signal is too faint in PD treated group on CAL27 cell line. And why is flow cytometry performed to determine the ratio of apoptotic and necrotic cells?
- In fig 6B,6C and 6D, why did the E-cadherin expression decrease in the 1mM treated groups?
- The paper did not test the toxicity of polydatin on normal cell lines. Thus, we do not know if PD can have high selectivity towards cancer cells.
- In line 220, the statement is not scientific sound and supported by your data. Caspase 3 activation does not induce release of cytochrome c, but the apoptosome formation can lead to the cleavage of caspase 3.
- In line 247, Ref 36 only conducted in vivo work and did not provide data to support the low endocytosis could be the reason for the high dose of PD here.
Author Response
- We appreciate your valuable comments, and additional experiments have been conducted for this paper to be published in 'Pharmaceutical'. (Results in Figures 1, 3 and 6)
Please see the attached file.

Reviewer 3 Report
I was not satisfied with the section of Introduction as well as Discussion.
1. The section of "Discussion" is suggested to be modified by providing some sub-titles similar to the sub-titles existing in the section of "Results."
2. In the last paragraph of the section of Introduction, write a sentence to clearly show the novelty of the current study.
3. Write a general paragraph about Cancer and its importance and mortality rates at the beginning of the section of Introduction. the following articles may contain innovative information to be cited in this paragraph. These articles include "Emerging theranostic silver and gold nanobiomaterials for breast cancer: Present status and future prospects (2021)"; "Emerging Theranostic Silver Nanomaterials to Combat Lung Cancer: A Systematic Review (2020)"; "Antineoplastic Biogenic Silver Nanomaterials to Combat Cervical Cancer: A Novel Approach in Cancer Therapeutics (2020)"
Author Response
- Thank you for increasing the value of this manuscript with your valuable attention and opinion.
- Please see the attached file.

Round 2
Reviewer 2 Report
Thanks for addressing the concerns. The manuscript has been greatly improved and can be considered to be accepted.
This manuscript is a resubmission of an earlier submission. The following is a list of the peer review reports and author responses from that submission.
Round 1
Reviewer 1 Report
The paper by Bang et al. describes the effect of polydatin on the viability, proliferation, apoptosis, autophagy, migration and invasion of OSCC cells.
Several important issues need to be addressed before the paper can be published.
- The Authors do not refer to several papers which described the effects of polydatin in HNSCC: Mele et al. https://www.nature.com/articles/s41419-018-0635-5 Martano et al. https://www.spandidos-publications.com/or/40/3/1435 Vijayalakshmi et al. https://www.mdpi.com/2076-3921/8/9/375/htm
- The concentrations of polydatin that were used in this study were very high. This is in contradiction to other studies (Mele et al. - 10-30 µM; Martano et al. - 10-30 µM). The concentrations which were used by the Authors in their current study cannot be achieved in vivo what makes all the findings biologically insignificant. How can the Authors comment on the discrepancies in the results (e.g. IC50 for CAL27 reported by Martano et al. is ~20 µM) and the biomedical significance of the effects of the applied concentrations of the compound? please, provide information about the absorption and pharmacokinetics of polydatin in humans or in animal models.
- The issue of DMSO concentration in cell culture has to be clarified. What was the concentration of DMSO in the medium?
- The Authors do not state clearly what controls were used in the experiments. Does "control" or "0" mean DMSO treated cells? On the other hand, in Figure 5 there is "Control" and "PD 0 mM" - what is the difference between these controls? The Authors must state unequivocally what type of controls were used in each experiment in the methodology section.
- Please, correct inaccuracies regarding cell death/apoptosis/autophagy in the text.
- Line 23 - should be 'fragmentation of procaspase-3'
- Line 116 - 'cleavage of procaspase-3'
- Line 24 - autophagy is not a form of apoptosis
- Line 50 - autophagy does not always lead to cell death, thus it cannot be considered 'another form of cell death'
- Lines 211-216 - please correct the flawed description of apoptosis, e.g. DNA fragmentation occurs after the release of cytochrome c and the activation of caspases
- Line 28 - please rephrase 'inhibiting cancer metastasis' because metastasis per se was not analyzed in this study (no in vivo experiments).
- Figure 1D - do the Authors have any ideas for the explanation of the drastic changes starting with 0.5 mM polydatin?
- Figure 2B - the results should rather be presented as '% of all cells' and not just the number of cells
- Figure 4 - A and B - in parallel with the photographs, quantitative data should also be presented in graphs
- Figure 5 - A - what is 'control'?, B - the Y axis states "width" while this is not in accord with the photographs - PD 0 mM should have lowest and not highest width, C - what is 'control'?
- In wound healing assay, what was the composition of the medium? What concentrations of FBS was used? were the cells FBS-starved before the start of the experiment?
- The time of treatment in cell invasion assay seems to be too long (72h) also because 24h treatments were used in other assays
- What was the exact procedure of data acquisition and analysis in the invasion assay? How were cells from the top chamber disposed of? How was the number of invading cells counted? [Lines 309-314 are too ambiguous].
- The concentration of 1 mM frequently stands out when it comes to effects since the effects start to fade (Fig. 3; 6B) - do the Authors have a possible explanation for this?
- Colony formation assay - what number of cells were seeded?
- Immunofluorescence (par. 4.5) - please provide information on the source of antibodies and chemicals.
- Line 43 - speling mistake 'stilbene'
- Line 89 - spelling mistake "Hoechst"
- Line 3 (title) - should be "...metastasis in oral squamous cell carcinoma cells in vitro"
Author Response
Response to Reviewer 1 Comments
ï‚· The Authors do not refer to several papers which described the effects of polydatin in HNSCC: Mele et al. https://www.nature.com/articles/s41419-018-0635-5 Martano et al. https://www.spandidos-publications.com/or/40/3/1435 Vijayalakshmi et al. https://www.mdpi.com/2076-3921/8/9/375/htm The concentrations of polydatin that were used in this study were very high. This is in contradiction to other studies (Mele et al. - 10-30 µM; Martano et al. - 10-30 µM). The concentrations which were used by the Authors in their current study cannot be achieved in vivo what makes all the findings biologically insignificant. How can the Authors comment on the discrepancies in the results (e.g. IC50 for CAL27 reported by Martano et al. is ~20 µM) and the biomedical significance of the effects of the applied concentrations of the compound? please, provide information about the absorption and pharmacokinetics of polydatin in humans or in animal models.
- We confirmed this part through repeated experiments several times. Experiments were conducted more than 5 times with the concentration listed above, and the material was checked more than 3 times (we also tried to change the manufacturer). However, at concentrations below 50 micromolar, it was confirmed that there was no change in either ca9-22 or CAL27. To confirm the question in this part, we are currently preparing an experiment to compare and apply different types of oral cancer cells and oral normal epithelial cells.
ï‚· The issue of DMSO concentration in cell culture has to be clarified. What was the concentration of DMSO in the medium?
- The cytotoxicity of DMSO in CAL27 and ca9-22 cells starts at 2%. In this study, up to 1% (v/v) DMSO was applied to cells.
ï‚· The Authors do not state clearly what controls were used in the experiments. Does "control" or "0" mean DMSO treated cells? On the other hand, in Figure 5 there is "Control" and "PD 0 mM" - what is the difference between these controls? The Authors must state unequivocally what type of controls were used in each experiment in the methodology section.
'Control' and 'PD 0mM' have not the same meaning, and all groups were dose not treated DMSO. Control in 5A refers to immediately after scratching the cells.
ï‚· Please, correct inaccuracies regarding cell death/apoptosis/autophagy in the text.
As per your advice, I revised the marked contents.
- Line 23 - should be 'fragmentation of procaspase-3'
- Line 116 - 'cleavage of procaspase-3'
- Line 24 - autophagy is not a form of apoptosis
- Line 50 - autophagy does not always lead to cell death, thus it cannot be considered 'another form of cell death'
However, there are many published studies that are classified as a type of cell death. Please consider the advantages.
- Lines 211-216 - please correct the flawed description of apoptosis, e.g. DNA fragmentation occurs after the release of cytochrome c and the activation of caspases
- When a cell enters the intrinsic pathway of apoptosis, the proteins present in the mitochondria are changed, and in particular, proteins such as cytochrome c are released into the cytoplasm to form a complex with the caspase protein present in the cytoplasm to undergo DNA fragmentation
ï‚· Line 28 - please rephrase 'inhibiting cancer metastasis' because metastasis per se was not analyzed in this study (no in vivo experiments).
- ultimately inhibiting cancer migration and invasion.
ï‚· Figure 1D - do the Authors have any ideas for the explanation of the drastic changes starting with 0.5 mM polydatin?
- In general, when checking cytotoxicity, it is based on that within 24 hours. The fact that there was a change at 0.5 mmol after 72 hours of polydatin treatment is a point that should also be considered with stress due to overgrowth of cells.
ï‚· Figure 2B - the results should rather be presented as '% of all cells' and not just the number of cells
- I revised it according to your advice.
ï‚· Figure 4 - A and B - in parallel with the photographs, quantitative data should also be presented in graphs
- AVO and autophaygosome were found in all cells(polydatin treated cells), so it is meaningless to express them graphically.
ï‚· Figure 5 - A - what is 'control'?, B - the Y axis states "width" while this is not in accord with the photographs - PD 0 mM should have lowest and not highest width, C - what is 'control'?
- Control in 5A refers to immediately after scratching the cells. The Y-axis of 5B means the area occupied by cells. It has been modified from width to area. Control in C means a group that is not treated with polydatin.
ï‚· In wound healing assay, what was the composition of the medium? What concentrations of FBS was used? were the cells FBS-starved before the start of the experiment?
- Experiments were carried out in normal medium containing 10% FBS.
ï‚· The time of treatment in cell invasion assay seems to be too long (72h) also because 24h treatments were used in other assays
- It was confirmed that even cells not treated with polydatin did not penetrate Metrigel until 48 hours. So we were able to obtain the results of Fig 5C by reducing the dose of Matrigel and increasing the incubation time.
ï‚· What was the exact procedure of data acquisition and analysis in the invasion assay? How were cells from the top chamber disposed of? How was the number of invading cells counted? [Lines 309-314 are too ambiguous].
- In the total area, the area of cells that penetrated the transwell and invaded on the membrane was digitized with photoshop and expressed as a graph.
ï‚· The concentration of 1 mM frequently stands out when it comes to effects since the effects start to fade (Fig. 3; 6B) - do the Authors have a possible explanation for this?
- I think that polydatin 1mM shows a definite apoptotic effect.
ï‚· Colony formation assay - what number of cells were seeded?
- 3*102 cells
ï‚· Immunofluorescence (par. 4.5) - please provide information on the source of antibodies and chemicals.
- Modified according to your advice..
ï‚· Line 43 - speling mistake 'stilbene'
- Modified according to your advice.
ï‚· Line 89 - spelling mistake "Hoechst"
- Modified according to your advice.
ï‚· Line 3 (title) - should be "...metastasis in oral squamous cell carcinoma cells in vitro"
- Modified according to your advice.

Reviewer 2 Report
I would like to appreciate your effort. Please find my comment as below:
As polydatin was extensively studies for many biological activities including anticancer activity with small dose (concentration) than the present study, so please ensure its benefits for this particular cancer cell lines by other experiments such as selectivity indexing comparing normal cell lines and relative to other cancer cell lines. Flow cytometry study to find out cell cycle information and apopercentage study.
Please also include how you prepare the polydatin during cell clines assay as this is highly polar compounds with poly -OH groups which cannot penetrate cell.
Author Response
I would like to appreciate your effort. Please find my comment as below:
As polydatin was extensively studies for many biological activities including anticancer activity with small dose (concentration) than the present study, so please ensure its benefits for this particular cancer cell lines by other experiments such as selectivity indexing comparing normal cell lines and relative to other cancer cell lines. Flow cytometry study to find out cell cycle information and popercentage study.
- Thank you for your valuable comments. We have confirmed that polydatin cytotoxicity was at rather high concentrations in oral cancer compared to other types of carcinoma. For this reason, we are currently preparing an experiment to compare polydatin to normal epithelial cells in the oral cavity and other types of oral cancer cells.
Please also include how you prepare the polydatin during cell clines assay as this is highly polar compounds with poly -OH groups which cannot penetrate cell.
- We have added corrections based on your comments.
Round 2
Reviewer 1 Report
The Authots did not address all the raised issues sufficiently. Especially, the Authors continue to ignore other papers on the same topic and do not refer to studies by Mele et al. https://www.nature.com/articles/s41419-018-0635-5 Martano et al. https://www.spandidos-publications.com/or/40/3/1435 Vijayalakshmi et al. https://www.mdpi.com/2076-3921/8/9/375/htm.
The concentrations which were used by the Authors in their study cannot be achieved in vivo what makes all the findings biologically insignificant. How can the Authors comment on the biomedical significance of the effects of the applied concentrations of the compound? The Authors did not provide information about the absorption and pharmacokinetics of polydatin in humans or in animal models.
The scratch wound healing assay should be performed under low FBS concentration to exclude effects related to cell proliferation. Moreover, it is advised that cells should be FBS-starved prior to performing the scratch. None of these conditions were applied in the experiments reported in this study thus the results cannot be considered as valid with respect to cell migration analysis.
The reported strong effects of polydatin on invasion can be to a large extent explained by its pro-apoptotic effects at 0.5 mM for 72h. Thus, it cannot be agreed that the Authors unequivocally proved the anti-invasion activity.
Figure 1A-C - should be "% of control" and not "% of cell viability"
Lines 21-23 - please, correct the sentence (grammatical/stylistic errors)
Reviewer 2 Report
Thank you for your response and revision.